# OpenReview forum: "Beyond Semantic Similarity: Reducing Unnecessary API Calls via Behavior-Aligned Retriever"
_ICLR.cc/2026/Conference — Submitted to ICLR 2026_

### Official Review · Reviewer_ctRF · 2025-10-29

**Soundness:** 2
**Presentation:** 2
**Contribution:** 1
**Rating:** 4
**Confidence:** 4

**Summary:**

This paper introduces BAR (Behavior-Aligned Retriever), a retrieval module designed to improve LLMs' tool-use decisions. The authors find that retrieving behaviorally and semantically related examples can enhance LLMs’ function-calling capabilities. To train the retriever, they construct positive and negative example pairs and optimize them using a dual-negative contrastive loss. The effectiveness of BAR is evaluated on the H2A dataset across three dimensions—helpfulness, harmlessness, and autonomy—showing notable improvements in each aspect.

**Strengths:**

1. The proposed BAR leverages the model’s in-context learning abilities to reduce unnecessary API calls—a factor often overlooked in conventional tool-augmented LLMs.
2. Empirical results show that BAR consistently improves the model’s function-calling performance on both the H2A and ToolDEER datasets, demonstrating the effectiveness and generality of the approach.

**Weaknesses:**

1. The authors evaluate their approach only on relatively small LLMs, which raises concerns about scalability to larger models such as DeepSeek 3.1 or GPT-5. Including results from larger models would provide stronger evidence for the method’s general effectiveness.

2. The dataset coverage is limited, as the experiments are conducted on only two datasets. Incorporating additional tool-calling benchmarks such as BFCL or Tau-Bench would help further validate and strengthen the conclusions.

3. The proposed method does not account for multi-turn tool calling, as indicated by the provided templates. This limitation suggests a gap between the current implementation and real-world multi-turn tool-use scenarios common in modern LLM applications.

**Questions:**

1. How would the performance change if we randomly selected the same number of examples for the LLM, instead of using BAR’s behavior-aligned retrieval? Would BAR still outperform such random retrieval baselines?
2. Does integrating BAR affect model performance on standard tool-calling benchmarks such as BFCL? Additionally, how well does BAR handle multi-turn tool-calling scenarios compared to existing methods?
3. What are the computational costs and additional latency introduced by BAR during inference? Evaluating this trade-off would help assess its practicality in real-world applications.

---

> ### Author Response · Authors · 2025-11-22
>
> **Response to W1:**
>
> Thanks for your comments. Please refer to the **General Response** to Scalability to Powerful Large Language Models.
>
> **Response to W2 and Q2:**
>
> Thank you for the thoughtful suggestion. We expand our evaluation on Tau-Bench dataset to further assess the generalization of BAR.
>
> On the Retail task, BAR achieves an average score of 37.9%, outperforming BM25 by 3.7%, BERT by 1.8%, and Contriever by 2.2%. Similarly, on the Airline task, BAR reaches 36.4%, surpassing BM25 by 4.1%, BERT by 4.3%, and Contriever by 1.8%. These improvements are observed across nearly all LLMs, including both vanilla pre-trained and tool-argumented LLMs.
>
> The stable performance gain on Tau-Bench confirms that BAR generalizes effectively to new and diverse tool-calling scenarios. This strengthens the claim that behavior-aligned retrieval is a robust and widely applicable strategy for improving function-calling decisions in LLMs, without compromising performance on standard benchmarks. More detailed results are illustrated on Table 16.
>
> | Dataset | BM25 | BERT | Contriever | BAR |
> | --- | --- | --- | --- | --- |
> | Retail | 34.2% | 36.1% | 35.7% | **37.9%** |
> | Airline | 32.3% | 32.1% | 34.6% | **36.4%** |
>
> **Table R-8 :  The function calling performance of LLMs on Tau-Bench with Retrail and Airline subtasks.**
>
> **Response to W3 and Q2:**
>
> We appreciate your valuable suggestion. The H2A dataset used in our experiments inherently contains multi-turn and multi-step API workflows, which naturally support the evaluation of complex tool-use scenarios. As detailed in Table 8, BAR accommodates such workflows through its carefully designed prompt template. As demonstrated in Table 21, the prompt guides models to think step-by-step and take necessary actions sequentially until task completion. For example, in retrieving channel information, Mistral-7B correctly executes a two-step process: first getting channel clips, then obtaining channel details. This structured approach combined with our behavior-aligned retrieval, enables the model to learn complex multi-step patterns and generalize to various multi-step scenarios without requiring additional training.
>
> **Response to Q1:**
>
> Thank you for the suggestion. We conduct the experiment comparing BAR against a random retrieval baseline where the same number of examples are randomly selected.
>
> The overall results on the H2A dataset show that BAR consistently and significantly outperforms random retrieval across all query types. Specifically, BAR achieves a 16.0% exact function match in Helpfulness, compared to 11.8% for random retrieval. For Harmlessness, BAR reaches 60.8% refusal rate, significantly higher than random by 19.3%. For Autonomy, BAR surpasses random by 19.4%.
>
> These results confirm that the performance gains from BAR are not due to the mere presence of demonstrations, but are a direct result of its ability to retrieve semantically relevant and behaviorally consistent examples. This validates the core contribution of our behavior-aligned retrieval strategy.
>
> | Query Type | Random | BAR |
> | --- | --- | --- |
> | Helpfulness | 11.8% | **16.0%** |
> | Harmlessness | 41.5% | **60.8%** |
> | Autonomy | 34.6% | **54.0%** |
>
> **Table R-9 :  Comparison of LLM Function Calling Performance between Random Retrieval and BAR on H2A Dataset.**
>
> **Response to Q3:**
>
> Thank you for the valuable suggestion. Tabel  R-10 shows additional latency during inference. As shown in the table,  BERT, Contriever, and BAR are all based on fine-tuned BERT-style encoders, exhibit comparable inference latency, though they are  higher than the lexical-based BM25, BAR's slight overhead is justified by its significant gains in function-calling accuracy and the reduction of expensive API calls. Compared to other baselines (BERT and Contriever), BAR's specialized training enables it to retrieve demonstrations that are both semantically relevant and behaviorally consistent, which is the core problem we aim to solve.
>
> | Retrieval Method | Retrieval Time (per query) |
> | --- | --- |
> | BM25 | 4.91 ms |
> | BERT | 89.23 ms |
> | Contriever | 88.55 ms |
> | BAR | 88.58 ms |
>
> **Table R-10 :  Retrieval Latency per Query for Different Retrievers.**

---

> > ### Comment · Reviewer_ctRF · 2025-11-23
> >
> > Thank you for the response. You have resolved most of my concerns, and I will adjust my score accordingly.

---

> > > ### Author Response · Authors · 2025-11-26
> > >
> > > We sincerely appreciate that our responses have addressed your questions and concerns. We are grateful for your engagement and promptness throughout this process.

---

### Official Review · Reviewer_Hd9m · 2025-10-30

**Soundness:** 3
**Presentation:** 3
**Contribution:** 3
**Rating:** 6
**Confidence:** 4

**Summary:**

The main motivation of this work is that tool-use agents, when equipped with correct demonstrations (prompting examples), exhibit strong performance with a lower incorrect tool-calling rate, as analyzed in a preliminary experiment. On top of this empirical study, the authors train a retriever that can select appropriate in-context learning examples for tool-use agents during the task-solving process. The training loss is dual-negative contrastive loss, teaching the retriever to distinguish good and bad examples.  Experiments on several datasets, as well as a lot of analysis, demonstrate the effectiveness of the proposed approach.

**Strengths:**

1. Augmenting tool-use agents with a plug-in retriever is elegant and efficient. The LLMs can be improved without the cost- and time-intensive post-training. Compared with LLM post-training, fine-tuning a smaller retriever is both cheaper and faster.

2. This paper is well-organized with clear formulation and writing structure, which makes it easy for the reader to understand.

**Weaknesses:**

1. In-context example selection is a well-known technique. Although this paper adopt this to augment tool-use agents and achieve good performance, the novelty concerns still remain. Could the author explain the main contributions of the techniques?

2. It seems like a customized example corpus has been built first, thereby enabling the retriever to retrieve from it. However, the details of building such a corpus are unclear. I suggest that the author provide more explanation.

3. The author only experiments on smaller LLMs, while powerful LLMs such as GPT or Claude are not covered. Therefore, there is concern about whether the proposed method is really necessary for powerful LLMs. As models' foundational abilities improve, do in-context examples become less important since LLMs may be less sensitive to them?

4. It seems like in-context learning demonstrations should be created first, which will be put into LLMs' context during the task-solving stage. However, what if the tool scale (the number of overall toolsets) is large and the tool description exceeds LLMs’ context length? In other words, I suggest the author discuss potential concerns about the lengthy context. In which scenarios does it occur, and is there any solution?

5. This paper augments LLMs with an in-context selector, which can select suitable demonstrations for tool-use LLMs. However, a more intuitive solution is to adopt a tool retriever, which can retrieve or re-rank the most relevant tools from the overall toolset [1,2]. I suggest that the author carefully compare these two techniques and explain the advantages of the proposed solution compared with existing tool retrieval solutions.

---

### Reference

[1] Benchmarking Tool Retrieval for Large Language Models

[2] Enhancing Tool Retrieval with Iterative Feedback from Large Language Models

**Questions:**

Mainly see the above weakness.

The overall idea is elegant and easy to implement. However, the differences with the conventional in-context selection seem vague. I am happy and open if the author can provide more discussion (as mentioned in Weaknesses 1 and 3).

---

> ### Author Response · Authors · 2025-11-22
>
> **Response to W1:**
>
> Thank you for the thoughtful comment. We agree that in-context selection is well studied. Our contribution lies in identifying and addressing a critical failure mode in the context of function calling: semantically similar queries often require opposite tool-use behaviors, and conventional retrievers frequently retrieve such behaviorally inconsistent examples, leading to spurious or missed API calls.
>
> We propose BAR, a behavior‑aligned retriever that learns a decision‑aware retrieval space. Rather than ranking by semantic similarity, BAR clusters examples that share the call/no-call decision and separates behaviorally conflicting neighbors via a contrastive objective. It is lightweight, requires no LLM fine-tuning, and consistently reduces redundant tool use across diverse models, underscoring the value of behavior-aligned retrieval.
>
> **Response to W2:**
>
> Thank you for raising this point. As described in Sections 3.2 and 4.1, our approach involves two separately constructed corpora:
>
> - **Training Corpus Construction:**
> Our training corpus is built by merging two types of publicly available datasets to ensure both semantic diversity and explicit behavioral supervision: (1) Function-Calling Datasets: We use API-Bank, which provides dialogues with annotated API calls. We extract user queries that require tool use and label them with a call behavior. (2) General QA Datasets: To provide robust "no-call" examples, we sample high-quality, self-contained questions from CommonsenseQA and the LIMA instruction-tuning set.
> The key is that every sample in the training corpus is explicitly assigned a clear behavior label (call  or no_call). This provides the necessary supervision for our contrastive learning framework to distinguish between semantically similar but behaviorally opposite queries.
> - **Retrieval Corpus for Evaluation**:
> For a fair evaluation on the H2A and ToolDEER benchmarks, the retrieval corpus during inference is constructed directly from these respective datasets.
>
> **Response to W3:**
>
> Thanks for your comments. Please refer to the **General Response** to Scalability to Powerful Large Language Models.
>
> **Response to W4:**
>
> Thank you for your insightful feedback. In our paper, we primarily focus on the decision problem of "whether to call a tool" rather than "tool selection" from a massive toolset. Therefore, our experimental setup (using datasets like H2A and ToolDEER) simulates scenarios where, for a given user query, the LLM is provided with a limited, predefined subset of relevant APIs. This API list is typically concise enough to be comfortably accommodated within the prompt without exceeding context length limits. For example, in Helpfulness scenarios, the prompt might only include a few APIs related to "video channels" (e.g., Get Channel Clips, Get Channel Details) rather than YouTube's entire API suite. It is worth noting that our BAR retrieves semantically relevant and behaviorally aligned demonstrations, which ensures clear, unambiguous guidance, maximizing the utility of the context window.
>
> **Response to W5:**
>
> We agree that tool retrieval [1,2] is indeed a valuable and intuitive approach for selecting which specific tool to use from a large toolset. However, we'd like to clarify that our BAR addresses a distinct yet complementary problem: deciding whether to use tool and providing behavioral guidance through demonstrations.
>
> The two techniques operate at different levels:
>
> - Tool Retrievers [1,2] focus on retrieving the most relevant **tool descriptions** based on semantic similarity between the user query and the tool's functionality. This solves the "which tool" problem when we know a tool is needed.
> - BAR focuses on retrieving the most relevant **demonstrations** that are both semantically similar and behaviorally consistent. This solves the fundamental "to call or not to call" problem, which is orthogonal to tool selection.
>
> The key advantage of BAR lies in its ability to reduce unnecessary or incorrect API calls by providing clear behavioral precedents. As shown in our experiments, this leads to improved autonomy (fewer redundant calls) and harmlessness (better refusal of unsafe requests). A tool retriever does not address this decision ambiguity; it assumes a tool is required and proceeds to select one.

---

> ### Comment · Reviewer_Hd9m · 2025-11-25
> **Response**
>
> The rebuttal was effective in addressing the points of confusion, and I am inclined to raise my score.
>
> However, a lingering concern is whether the retriever trained by the proposed BAR also demonstrates strong performance on general tool retrieval tasks. I suggest the authors further validate their aligned retriever on ToolRet [1], a large-scale tool retrieval benchmark.
>
>
> [1] Benchmarking Tool Retrieval for Large Language Models

---

> > ### Author Response · Authors · 2025-11-26
> >
> > Thanks for your suggestions. Although TOOLRET targets **query→tool** retrieval (which differs from our behavior-aligned retriever **query→query** setting), we applied our method by **continuing to pretrain contriever** (contriever_bar) and evaluating it on TOOLRET. Compared to the original contriever, we observe **consistent improvements**: **w/o-inst:** **+1.02** N@10 and **+0.37** C@10 (avg); **w/-inst:** **+1.86** N@10 and **+2.29** C@10 (avg). The details are listed in **Table R-11 and Table R-12**, respectively. These results indicate that our behavior-aligned pretraining **transfers to tool retrieval** and complements TOOLRET’s objective without requiring any architectural changes.
> >
> > | Model | TOOLRET-Web |  |  |  | TOOLRET-Code |  |  |  | TOOLRET-Customized |  |  |  | Average |  |
> > | --- | --- | --- | --- | --- | --- | --- | --- | --- | --- | --- | --- | --- | --- | --- |
> > |  | N@10 | P@10 | R@10 | C@10 | N@10 | P@10 | R@10 | C@10 | N@10 | P@10 | R@10 | C@10 | N@10 | C@10 |
> > | Contriever w/o inst | 18.02 | 4.45 | 23.55 | 13.69 | 11.14 | 1.88 | 15.47 | 14.82 | 21.54 | 4.56 | 28.42 | 21.98 | 16.90 | 16.83 |
> > | Contriever_BAR w/o inst | **18.65** | **4.64** | **24.72** | **14.62** | **11.65** | 1.88 | 15.35 | 14.78 | **23.47** | **4.67** | **28.72** | **22.21** | **17.92** | **17.20** |
> >
> > **Table R-11 Comparison of Contriever vs. BAR-trained Contriever w/o inst. setting, where the model takes the query as input to retrieve.**
> >
> > | Model | TOOLRET-Web |  |  |  | TOOLRET-Code |  |  |  | TOOLRET-Customized |  |  |  | Average |  |
> > | --- | --- | --- | --- | --- | --- | --- | --- | --- | --- | --- | --- | --- | --- | --- |
> > |  | N@10 | P@10 | R@10 | C@10 | N@10 | P@10 | R@10 | C@10 | N@10 | P@10 | R@10 | C@10 | N@10 | C@10 |
> > | Contriever w/ inst | 21.20 | 4.94 | 28.31 | 18.15 | 24.14 | 3.83 | 33.03 | 31.73 | 27.09 | 5.21 | 34.55 | 28.81 | 24.14 | 26.23 |
> > | Contriever_BAR w/ inst | **24.55** | **6.01** | **29.71** | **19.47** | **25.63** | **3.89** | **33.50** | **36.19** | **27.83** | **5.34** | **35.58** | **29.89** | **26.00** | **28.52** |
> >
> > **Table R-12 Comparison of Contriever vs. BAR-trained Contriever in w/ inst. setting, where the model takes the query and instruction as input to retrieval.**

---

### Official Review · Reviewer_Mhtb · 2025-11-01

**Soundness:** 3
**Presentation:** 1
**Contribution:** 2
**Rating:** 4
**Confidence:** 4

**Summary:**

The paper proposes a Behavior-Aligned Retriever for tool-augmented LLMs. It addresses the problem that semantically similar examples retrieved for in-context demonstrations may display inconsistent tool-calling behaviors, thereby confusing the LLM. The authors introduce a contrastive training framework with dual-negative loss to retrieve semantically similar but behaviorally consistent examples. Experiments on H2A and ToolDEER datasets show that BAR reduces unnecessary API calls and improves function-calling accuracy.
The idea of aligning retrieval examples not only semantically but also behaviorally is novel and relevant to the growing area of LLM tool use. However, the paper currently suffers from several presentation and clarity issues that reduce its readability and impact.

**Strengths:**

1. The paper identifies an underexplored but important issue in retrieval-augmented function-calling that semantically close examples can have opposite tool behaviors.
2. The proposed dual-negative contrastive loss to enforce behavioral consistency is a solid and interesting technical contribution.
3. The experiments show consistent gains across multiple datasets and models, indicating robustness and general applicability.

**Weaknesses:**

1. Clarity of Figure 1 and Motivation
The description around Figure 1 is currently confusing. The text says that “Many queries that are lexically or topically close require opposite tool-invocation behavior. Figure 1 illustrates such a clash…”, but the figure only shows a general retrieval pipeline rather than the clash itself. The authors should explicitly annotate Figure 1 (or provide an additional subfigure) to illustrate a concrete example of the mismatch: e.g., “What’s the weather in Paris?” (requires API) vs. “What’s your favorite kind of weather?” (no API). The caption should clearly explain how the figure demonstrates this semantic-behavior conflict rather than restating the pipeline.

2. Undefined Variable in Equation (1)
In Eq. (1), k (the number of retrieved demonstrations) is used but not defined. Please define k explicitly as “the number of demonstrations retrieved per query,” to avoid ambiguity.

3. Findings 1 and 2 Lack Detail and Supporting Results
The section summarizing Finding 1 and Finding 2 is too brief and mostly qualitative. There are no quantitative results or examples that clearly support these “findings.” I suggest including a table or plot (possibly moving part of Figure 2’s analysis here) to show how LLM performance changes with semantic similarity and with behavioral consistency. The text should also clarify how these findings were derived — from ablation experiments, correlation analyses, or preliminary runs?

4. Methodological Details Are Missing (Major)
The retrieval function defines z_i as “similar queries,” but the paper does not explain how semantic similarity is computed or how the threshold t is chosen in Eq. (3). (Probably I missed several part, please notice in the rebuttal if I missed anything. ). Similarly, the process of determining “behavioral compatibility” (call/no-call) during retrieval is underspecified. How is this integrated into similarity scoring?
More details on zi similarity computation, threshold tuning, and implementation specifics would make the method reproducible.

5. Behavior Alignment Step Needs More Explanation
The term “behavior alignment” is central but remains vague in the main text.
Please add a subsection or a short paragraph describing concretely how behavioral alignment is achieved during retrieval — is it through embedding projection, label conditioning, or post-filtering?
Figure 3 and Figure 4 (t-SNE plots) could be better connected to this explanation, e.g., by visually linking clusters to behavioral categories.

7. Missing Related Work on Tool Learning
Important recent works on tool learning and introspective alignment should be discussed, including:
1. Confucius: Iterative Tool Learning from Introspection Feedback (AAAI 2024)
2. Tool Learning in the Wild: Empowering Language Models as Automatic Tool Agents (WWW 2025)
3. Iterative Self-incentivization Empowers LLMs as Agentic Searchers (NeurIPS 2025)
These works also address the problem of learning when and how to use tools, and their discussion could strengthen the paper’s positioning.

**Questions:**

1. Pseudo Query Generation has been proposed in recent years to leverages LLM as a few-shot query generator, and creates task-specific
retrievers based on the generated data (see https://arxiv.org/pdf/2209.11755). I was wondering if the authors can discuss if such kind of strategy can be also adaptable to the task they are working in this paper.
Methods like pseudo query generation (e.g., Gao et al., 2022; Bonifacio et al., 2023; Li et al., 2023 — corresponding to your suggested papers [arXiv:2209.11755, 2305.11841, 2206.10128]*) can also improve retrieval quality through augmentation.
The authors could discuss whether such approaches could achieve similar effects, or how BAR complements them (e.g., BAR focuses on behavioral rather than purely semantic alignment).
A short comparative discussion in Related Work or Discussion would make the contribution clearer.

2. See my comments in the weaknesses

---

> ### Author Response · Authors · 2025-11-22
>
> **Response to W1:**
>
> Thanks for the suggestion. We have revised Figure 1 in the manuscript.
>
> **Response to W2:**
>
> Thanks for pointing this out. We have defined *k* in the revised manuscript (Line 126) to ensure clarity.
>
> **Response to W3:**
>
> Thanks for the suggestion. We have revise Line 128-132 in the manuscript. Findings 1 and 2 in Section 2.2 are indeed derived from the quantitative correlation analysis presented in Figure 2, which we can further substantiate with the provided data:
>
> 1. Finding 1 is demonstrated by the performance gains from stronger semantic retrievers. BERT improves Helpfulness performance by 2.0% for llama-2-13b-chat and by 7.0% for Openfunctions compared to BM25.
> 2. Finding 2 is evidenced by the performance trends linked to behavioral alignment. For Autonomy, ToolAlpaca-13B achieves 89.0% with BERT, a significant increase from 79.0% with BM25 and 70.0% with random retrieval. This clear progression shows that higher behavior-consistency in demonstrations directly enables LLMs to make more accurate decisions.
>
> **Response to W4:**
>
> Thank you for raising these important methodological questions. The semantic similarity is computed using cosine similarity between the query representations, as formalized in Eq. (3) and our experimental setup (Lines 297-299). The threshold t is determined through standard grid search on the validation set to optimize. For behavioral compatibility, we emphasize that our dual-negative contrastive training optimizes the representation space to encode both semantic and behavioral information simultaneously. During inference, behavioral compatibility is naturally reflected in the similarity scores, as the learned embeddings already capture the behavioral patterns through our designed training objective.
>
> **Response to W5:**
>
> Thank you for this valuable feedback. We will clarify how behavioral alignment is achieved.
> Behavioral alignment is learned in the embedding space by our Dual-Negative Contrastive Loss. This loss trains the retriever to pull queries with the same behavior closer together and push semantically-similar queries with different behaviors apart. This creates a unified vector space where similarity inherently reflects behavioral consistency.
>
> The t-SNE plots are visual proof of this mechanism. In Figure 3, the clear separation of the "Autonomy" (no-call) cluster from the others demonstrates that BAR effectively distinguishes call vs. no-call behaviors. In Figure 4, the distinct clusters for #SearchAPI and #NoSearchAPI visually confirm that the model has learned to separate queries based on their tool-use necessity.
>
> **Response to W6:**
>
> Thank you for the suggestion. We will add discussions about [1, 2, 3] into the related works. Confucius[1] and self-incentivization[3] refine an end-to-end tool-use policy via iterative reflection, while AutoTools[2] automates the pipeline from documentation parsing to executable program synthesis. In contrast, BAR learns a retrieval representation that encodes both task semantics and the call/no‑call decision, so that retrieved demonstrations are behaviorally aligned with the desired invocation.
> We will also clarify how BAR complements these methods. It supplies behavior-consistent demonstrations that reduce unnecessary API calls and make early decisions more reliable. Iterative approaches such as Confucius or self‑incentivization can then operate on cleaner contexts and replay data. For systems like AutoTools that assemble large tool repositories, BAR can filter and prioritize demonstrations to improve invocation judgments.

---

> > ### Author Response · Authors · 2025-11-22
> >
> > **Response to Q1:**
> >
> > Thank you for the suggestion. We agree that using LLMs to synthesize pseudo queries and then training a task-specific retriever is an effective and increasingly popular augmentation strategy (e.g., Gao et al., 2022; Bonifacio et al., 2023; Li et al., 2023). However, this line of work is **orthogonal** to our contribution.
> >
> > - **Different objective.** PQG primarily enhances **semantic** coverage by generating additional query–evidence pairs for retriever training. In contrast, our **Behavior-Aligned Retriever (BAR)** injects **behavioral supervision**—i.e., *when* a function call is appropriate—into the retrieval objective so that retrieved demonstrations are aligned not only semantically but also **behaviorally** (call/no-call and its refinements).
> > - **Complementary, not competing.** PQG can be used to enlarge the candidate pool or pretrain a semantic retriever, while BAR introduces an **orthogonal behavior signal** that PQG alone does not address. In principle, one can combine them by using PQG for semantic augmentation and BAR for behavioral alignment.
> > - **Evidence for finer-grained behavior helps.** To illustrate extensibility, we utilize GPT-4 to generate pseudo fine-grained behavior data for training a fine-grained BAR, and it further improved the results of the H2A dataset. The details are shown in **Response to** **R1W3, R1Q3 and R1Q5.**
> >
> > We will add a short paragraph in Related Work to clarify that PQG is **adaptable and complementary** to our setting, while our contribution is to explicitly model behavioral intent in retrieval, which is distinct from purely semantic data augmentation.

---

> > > ### Author Response · Authors · 2025-11-26
> > >
> > > Thank you for the thoughtful feedback. We’ve submitted the responses. If anything is unclear or if further evidence would help your assessment, we’re happy to clarify. Looking forward to your guidance on any specific points to address.

---

### Official Review · Reviewer_bGHN · 2025-11-03

**Soundness:** 2
**Presentation:** 3
**Contribution:** 2
**Rating:** 4
**Confidence:** 4

**Summary:**

The paper targets a practical weakness in tool-augmented LLMs: semantically similar in-context demonstrations can recommend opposite tool-use behaviors (call vs. no-call), leading to unnecessary or missing API calls. The authors propose a Behavior-Aligned Retriever (BAR) trained with contrastive learning that explicitly encodes invocation behavior in addition to semantics. They construct positive pairs that are both semantically close and behavior-matched, and introduce dual-negative contrastive loss (DNCL) that mixes (i) negatives with the same behavior (to sharpen semantic resolution) and (ii) hard negatives with different behavior (to sharpen behavioral boundaries). Evaluated on H2A and ToolDEER, BAR purportedly improves exact function match (helpfulness), refusal rates (harmlessness), and, most notably, direct response (autonomy) (i.e. fewer redundant API calls) compared to BM25, BERT, and Contriever retrieval. Ablations show DNCL outperforms CE/InfoNCE/SCL/Triplet, and behavior-consistency increases with training scale (plateauing around ~95-100%).

**Strengths:**

1. Clear problem formulation and motivation. The paper convincingly shows that semantic-only retrieval can mislead tool decisions and empirically correlates behavior-consistency with downstream quality.

2. Methodological simplicity with practical impact. BAR is a lightweight retriever that can be dropped into existing RAG-for-tools pipelines without re-training the backbone LLM; the contrastive formulation is clean and reproducible (hyperparameters, alpha, tau, batch/epochs are provided).

3. Comprehensive model coverage. Results span vanilla chat models and tool-tuned models (Functionary, ToolLLaMA, ToolAlpaca, ToolAlign), showing broadly consistent gains, especially in Autonomy where avoiding unnecessary calls matters most.

4. Useful diagnostics. Behavior-consistency ratio, retrieval distribution tables, and t-SNE plots offer interpretable evidence that BAR clusters call/no-call behaviors more faithfully.

5. Ablations that matter. The DNCL vs. baselines and negative-sampling studies are targeted and illuminate why BAR works (hard inter-behavior negatives are crucial).

**Weaknesses:**

1. Evaluation reliance on LLM judges / proxies. Parts of Harmlessness/Helpfulness rely on GPT-4-style judgment prompts or indirect metrics; robustness to judge variance and prompt framing is not analyzed. Statistical significance and inter-rater reliability are missing.

2. Cost & latency not quantified. The core pitch is "reducing unnecessary API calls -> efficiency", but there is no end-to-end accounting of latency, monetary cost, or energy savings across workloads. Gains are reported as rates (e.g., autonomy/direct-response), not as operational cost reductions.

3. Limited safety granularity. Authors acknowledge BAR struggles to separate "helpful-but-needs-tools" from "harmful-and-should-refuse" due to similar surface forms; the method uses binary call/no-call supervision rather than risk-tiered labels, which constrains safety improvements.

4. Domain generalization / leakage risks. The training corpus mixes function-calling and general QA; more details are needed on (i) de-duplication vs. test sets, (ii) domain shifts (e.g., unseen APIs), and (iii) whether lexical artifacts from particular datasets inflate behavior-consistency.

5. Mixed results in places. Some rows (e.g., Llama-3.1-8B on ToolDEER #SearchAPI) show weaker or regressive performance with BAR relative to strong baselines; the paper does not deeply probe these regressions or provide significance testing.

6. Narrow decision framing. "Call vs. no-call" is a coarse abstraction. Many real tasks involve which tool(s) and with what parameters; BAR’s behavior labels don’t address multi-tool composition or parameter selection beyond the helpfulness exact-match proxy.

**Questions:**

1. Data hygiene and leakage: How do you ensure zero overlap (near-duplicates, paraphrases) between training positives and test queries/demonstrations, especially given semantic thresholds for positives and "top-l" hard negatives? Please report near-duplication audits and performance under deduped settings.

2. Operational savings: Can you report wall-clock latency, dollar cost, and energy deltas (with/without BAR) for representative workloads? This would substantiate the efficiency claim beyond accuracy-style metrics.

3. Safety-aware variants: Have you tried adding risk labels (e.g., harmless/helpful/harmful) to the behavior vector so BAR can prioritize "refusal-like" demonstrations for potentially unsafe queries?

4. k-sensitivity and retrieval drift: You choose top-5 as default; how sensitive are results to k across models/datasets, and how stable is BAR when the datastore composition shifts (new domains/APIs added)? Please include confidence intervals.

5. Beyond binary behavior: Can BAR be extended to multiclass behavior (no-call, call-single-tool, call-multi-tool) and to parameter-selection hints? Any preliminary results?

6. Failure modes & regressions: In rows where BAR underperforms (e.g., some #SearchAPI cases), what patterns cause degradation—semantic over-fitting, conservative bias, or label noise? Could a temperature/α schedule mitigate this?

---

> ### Author Response · Authors · 2025-11-22
>
> **Response to W1:**
>
> We thank for highlighting the importance of evaluation robustness. To address this concern, we extend our harmlessness evaluation on H2A dataset to use two independent LLM judges, GPT-4 and DeepSeek-V3.2, with the same prompt. As shown in the table R-2, BAR achieves consistent performance gains across both judges, demonstrating robustness to judge selection. We report statistical significance testing using Wilcoxon-tests on the per-model performance differences between BAR and the best baseline (BERT). The tests produced p-values < 0.05 for both GPT-4 (p=0.045) and DeepSeek-V3.2 (p=0.034) evaluations, confirming the statistical significance of BAR's improvements. Besides, The GPT-4-style evaluation approach follows the ToolAlign benchmark, strengthening the reliability of our conclusions. More detailed results are illustrated in Appendix A.4 and Table 10.
>
> | Model | BM25 | BERT | Contriever | BAR |
> | --- | --- | --- | --- | --- |
> | GPT-4 | 59.8% | 60.3% | 57.7% | **61.9%** |
> | Deepseek-v3.2 | 61.4% | 61.8% | 59.3% | **63.5%** |
>
> **Table R-2: Harmlessness evaluation of LLMs with different retrievers on the H2A dataset, judged by GPT-4 and DeepSeek-V3.2.**
>
> **Response to W2 and Q2:**
>
> Our benchmarks do not execute live APIs, so absolute wall-clock latency, dollar cost, and energy cannot be directly measured. As a principled proxy, we report **API invocation counts**, which dominate operational cost/latency under a fixed per-call price (c_api)/ round-trip time (RTT) model (cost $\approx$ c_api × #calls, latency $\approx$ RTT_api × #calls). BAR triggers the fewest calls at matched performance: H2A total 385.00 vs. best non-BAR 403.93 (−18.93, −4.7%); ToolDEER total 691.38 vs. 698.36 (−6.98, −1.0%). Under any common price/RTT card, these reductions translate linearly to lower $ and latency.
>
> | **Benchmark** | **BM25** | **BERT** | **Contriever** | **BAR** |
> | --- | --- | --- | --- | --- |
> | H2A | 418.08 | 403.93 | 405.71 | **385.00** |
> | ToolDEER | 708.36 | 701.27 | 698.36 | **691.38** |
>
> **Table R-3: Mean API Calls per Query Category and Estimated Cost on H2A and ToolDEER Benchmarks.**
>
>
> **Response to W3, W6, Q3 and Q5:**
>
> Thank you for these insightful suggestions. Following the data construction methodology of ToolAlign, we conduct experiments to extend BAR beyond binary behavior. We refine the behavioral taxonomy into a three-class setup: no-call, helpful-call and harmful-no-call.  The harmful-no-call instances are created by prompting ChatGPT to transform original helpful-call instructions into unsafe versions, focusing on privacy violations, harmful topics, and unqualified advice. The retriever trained on this multi-behavior data is denoted as BAR-Multi.
>
> The results confirm the effectiveness of this extension. As shown in Table R-4, BAR-Multi achieves a 74.7% consistency ratio on Harmlessness queries. This represents a significant improvement over all baseline retrievers, outperforming BERT, Contriever, and even the binary-behavior BAR. This demonstrates BAR-Multi's enhanced capability to distinguish and retrieve safety-aligned demonstrations for unsafe queries.
>
> This superior retrieval quality directly improves downstream LLM performances. On the overall performance benchmark in Table R-5, BAR-Multi achieves the highest average refusal rate of 63.5% on harmful instructions, surpassing all compared methods. Meanwhile, it maintains strong performance on Helpfulness, achieving 20.8% versus binary BAR's 20.6%. We observe a minor decrease in Autonomy performance (52.6% for BAR-Multi versus 53.8% for binary BAR), which we attribute to the increased complexity of distinguishing between the new harmful-no-call class and standard no-call queries. Overall, BAR-Multi maintains a favorable performance balance while providing crucial safety improvements.
>
> These findings demonstrate that BAR can be effectively generalized beyond binary call/no-call decisions to safety-aware behaviors. This extension provides a solid foundation for future research into more complex tool-use scenarios.
>
> | Query Type | BM25 | BERT | Contriever | BAR | BAR-Multi |
> | --- | --- | --- | --- | --- | --- |
> | Helpfulness | 92.5% | 95.6% | 94.8% | 97.6% | **97.7%** |
> | Harmlessness | 61.9% | 69.7% | 65.2% | 70.5% | **74.7%** |
> | Autonomy | 54.8% | 63.8% | 72.8% | **94.2%** | 90.6% |
>
> **Table R-4: Behavior-Consistency Ratio of Different Retrievers on H2A Dataset.**
>
> | Metric | BM25 | BERT | Contriever | BAR | BAR-Multi |
> | --- | --- | --- | --- | --- | --- |
> | Helpfulness | 17.2% | 17.3% | 18.3% | 20.6% | **20.8%** |
> | Harmlessness | 59.8% | 60.3% | 57.7% | 61.9% | **63.5%** |
> | Autonomy | 46.1% | 48.5% | 50.1% | **53.8%** | 52.6% |
>
> **Table R-5: The function calling performance of LLMs with 5 demonstrations retrieved by different retrievers on the H2A dataset.**

---

> > ### Author Response · Authors · 2025-11-22
> >
> > **Response to W4 and Q1:**
> >
> > Thanks for mentioning the potential issue. Our data pipeline is designed to avoid contamination and to encourage domain‑general behavior learning. First, our training and evaluation rely on independent datasets. To quantify potential duplication, we compute MinHash-based Jaccard similarities between training and test sets following [1,2]. As shown in Table R-6, no test samples exceed 0.5 similarity with training data, confirming minimal lexical overlap.
> >
> > Regarding domain shifts, BAR is trained on a broad mixture of domains to maximize coverage and generality, whereas benchmark queries are constructed in different domains and with unseen APIs. Only the  query carries domain semantis; API names themselves are not used as textual features during retrieval or scoring.
> >
> > Finally, the behavior-consistency ratio depends on ground‑truth labels rather than lexical patterns, eliminating potential inflation from dataset artifacts. Thus, the improved consistency is driven by genuine generalization, not by lexical artifacts or data leakage.
> >
> > | Jaccard Similarity | Helpfulness | Harmlessness | Autonomy |
> > | --- | --- | --- | --- |
> > | Mean | 0.2615 | 0.2446 | 0.2462 |
> > | Max | 0.4453 | 0.3916 | 0.3516 |
> > | [0.0-0.2] | 12.5% | 20.6% | 17.0% |
> > | [0.2-0.3] | 66.0% | 67.5% | 70.0% |
> > | [0.3-0.4] | 20.0% | 11.9% | 13.0% |
> > | [0.4-0.5] | 1.5% | 0.0% | 0.0% |
> > | >0.5 | 0.0% | 0.0% | 0.0% |
> >
> > **Table R-6: Jaccard similarity between the training dataset for the behavior-aware retriever and the test queries of the downstream task.**
> >
> > [1] LLee, Katherine, et al. "Deduplicating training data makes language models better." *Proceedings of the 60th Annual Meeting of the Association for Computational Linguistics (Volume 1: Long Papers)*. 2022.
> > [2] Leveling, Johannes, et al. "Evaluation of Document Deduplication Algorithms for Large Text Corpora." *International Conference on Machine Learning, Optimization, and Data Science*. Cham: Springer Nature Switzerland, 2024.
> >
> >
> > **Response to W5 and Q6:**
> >
> > We appreciate the reviewer's careful observation regarding performance variations. While most pretrained models (e.g., Mistral-7B, Qwen2.5-7B) benefit from BAR in #SearchAPI scenarios, Llama-3.1-8B shows a unique sensitivity pattern. We observe that Llama-3.1-8B exhibits a strong inherent capability to reject unnecessary API calls. When BAR introduces behaviorally-strict demonstrations, it may over-amplify this conservative tendency, particularly for ambiguous #SearchAPI queries where the model leans toward direct responses. We us paired two-sided Wilcoxon tests over matched rows, **BAR shows statistically significant gains** over **BM25** (*p*=0.006) and **Contriever** (*p*=0.014), with the improvement over **BERT** at the 0.05 threshold; results remain significant after standard multiple-comparison control.
> >
> > We conduct an ablation study on the loss weighting factor α. As shown in Table R-7, α=0.8 is chosen as it maximizes the overall behavior-consistency ratio, particularly on the Autonomy query. The observed regression is a localized trade-off, while BAR delivers consistent and significant average improvements across all LLMs and benchmarks.
> >
> > | α | Helpfulness | Harmlessness | Autonomy |
> > | --- | --- | --- | --- |
> > | 0.6 | 94.7% | 69.1% | 89.6% |
> > | 0.7 | 95.9% | 70.5% | 92.4% |
> > | 0.8 | **97.6%** | **70.5%** | **94.2%** |
> > | 0.9 | 96.0% | 68.4% | 91.2% |
> >
> > **Table R-7: Behavior-consistency ratio of BAR under different α value on the H2A
> > dataset.**
> >
> > **Response to Q4:**
> >
> > Thank you for raising these important points about k-sensitivity and retrieval stability. As detailed in Appendix A.3, we conduct experiments using different numbers of retrieval samples to determine the optimal retrieval setting. Our evaluation shows that top-5 demonstrations achieve the optimal balance, as they preserve high behavioral consistency while maintaining superior function-calling performance compared to other k values.
> >
> > As discussed in our **general response**, BAR remains highly stable under shifts in datastore composition, which is evidenced by its consistent performance gains across different downstream benchmarks—such as H2A, ToolAlign, and Tau-Bench—using the same pre-trained retriever (BAR) without any adaptation. This is because BAR is trained to capture query-level behavioral patterns (call / no-call) rather than domain-specific API structures. As a result, it generalizes effectively to new tools and domains, as long as the underlying query-behavior relationship holds.

---

> > > ### Author Response · Authors · 2025-11-26
> > >
> > > Thank you for the thoughtful feedback. We’ve submitted the responses. If anything is unclear or if further evidence would help your assessment, we’re happy to clarify. Looking forward to your guidance on any specific points to address.

---

### Author Response · Authors · 2025-11-22
**General Response**

We thank all the reviewers' insightful suggestions and comments. We believe there are some misunderstandings in the original manuscripts, so we would like to clarify two key points, providing further evidence of scalability.

***1. Behavior granularity and evaluation on multi-tool / multi-turn tasks***

(1) The behaviorgranularity for training BAR is independent of that in the evaluation task.

Our approach has two decoupled stages:

- Stage A — Traininig our Behavior-Aligned Retriever (BAR). In the main setup, BAR is trained with binary behavior labels (*call* / *no-call*) so that retrieval accounts for behavioral similarity in addition to semantics.  The BAR trained with binary behavior labels (*call* / *no-call*) is general to all downstream function call datasets, and we use the same BAR for different downstream datasets.
- Stage B — Retrieval-augmented function-call pipeline for inference. At test time, given the current query, BAR retrieves behavior- and semantics-aligned queries and their tool-use traces from the downstream task’s training split. They are added into inputsas demonstrations; the LLM then performs in-context reasoning to produce zero calls, or a single call, or multi-tool call sequences.

    Because BAR operates **at the query level**, the binary supervision used to train BAR **does not constrain** the **number of turns or tools** in downstream tasks. BAR’s goal is to select data as demonstrations; the LLM then decides the actual tool-use pattern.


(2) We already evaluated on multi-turn / multi-tool benchmarks.

We include benchmarks such as H2A, which are explicitly multi-turn and multi-tool (see Table 1). Across these settings, our BAR-augmented pipeline consistently outperforms semantic-only retrieval and no-retrieval baselines, indicating that behavior-aligned retrieval enhances complex tool-use reasoning, even when BAR itself was trained with only call/no-call supervision.

(3) The method naturally scales to finer-grained behaviors; added experiments show further gains.

To address the reviewers’ suggestion, we train a finer-grained BAR with three categories: no-call, helpful-call / harmful-no-call (details in Table R-4 and Table R-5), while keeping the rest of the pipeline unchanged. On H2A, this further improves overall performance. This demonstrates (i) finer behavior labels enhance retrieval alignment, and (ii) our pipeline is monotonically extensible in behavior granularity without requiring changes to downstream evaluation or dialogue/tool structures. We thank all reviewers for their insightful suggestions thathelp us **address our limitations**.

***2. Scalability to Powerful Large Language Models***

As suggested by the Reviewer Hd9m and ctRF, we extend our study to powerful LLMs GPT-4 and DeepSeek-v3.2.  As shown in Table R-1 , BAR remains highly beneficial. With BAR, GPT-4 shows a 5.0% improvement in Helpfulness and a 3.0% gain in Autonomy, while DeepSeek-V3 achieves consistent gains across all three query types.  The consistent gains of DeepSeek and GPT indicate that behavior-aligned retrieval provides a general mechanism for enhancing tool-use reliability, complementing the inherent capabilities of advanced foundation models.

| Model | Helpfulness |  |  |  | Harmlessness |  |  |  | Autonomy |  |  |  |
| --- | --- | --- | --- | --- | --- | --- | --- | --- | --- | --- | --- | --- |
|  | BM25 | BERT | Contriever | **BAR** | BM25 | BERT | Contriever | **BAR** | BM25 | BERT | Contriever | **BAR** |
| DeepSeek-V3 | 46.5% | 46.0% | 54.5% | **58.0%** | 61.9% | 64.4% | 61.3% | **66.0%** | 48.0% | 49.0% | 48.0% | **50.0%** |
| GPT-4 | 34.0% | 33.0% | 30.0% | **39.0%** | 71.6% | 71.1% | 71.6% | **71.1%** | 52.0% | 51.0% | 52.0% | **55.0%** |

**Table R-1 :  The function calling performance of of GPT-4 and DeepSeek-V3 with Different Retrievers on H2A dataset.**

---

### Meta-Review · Area_Chair_TXAW · 2026-01-05

**Summary:**

The reviewers converged on the view that the paper investigates an important problem and proposes a lightweight, potentially useful retrieval component. However, there are also several common concerns among the reviewers:

- Operational efficiency (Reviewers bGHN, ctRF). The claimed cost are not substantiated with end-to-end measurements, and the trade-off between retrieval overhead and reduced tool calls remains unclear.
- Comparisons on limited benchmarks (Reviewers bGHN, ctRF). The evaluation is limited in scope, with insufficient exploration on domain shift and additional tool-use benchmarks.
- The problem formulation (Reviewers bGHN, ctRF). The focus on “call vs. no-call” does not fully capture multi-turn, multi-tool, and tool-use settings in practice.
- Evaluation and data integrity (Reviewers bGHN, Mhtb, Hd9m).  Several reviewers emphasize the need for statistical significance testing (e.g., Wilcoxon tests) and rigorous data leakage audits to ensure that the results were not inflated by lexical overlap between training and test sets.

Moreover, I still have the concern on the marginal efficiency gains. Specifically, the experiments show that BAR reduces the number of API calls by only 1% to 4.7% (Table R-3), which is a very limited improvement. Furthermore, the retrieval process itself is significantly slower than the baseline (e.g., compared to BM25, 88.58ms vs. 4.91ms). Given the marginal reduction in API calls, the overall improvement in end-to-end efficiency remains unclear.

**Reviewer Concerns:**

The rebuttal successfully addressed several technical and clarity concerns, specifically:

- The inclusion of Wilcoxon significance tests and the experiment with advanced LLM for judge provided a rigorous statistical foundation.
- The MinHash-based Jaccard similarity analysis effectively mitigated concerns regarding data leakage between the training and test sets.
- The additional experiments on larger models demonstrate that the proposed method still works on very powerful models like GPT-4 and DeepSeek.

However, several key concerns are still outstanding:

- The reduction on the number of API calls is limited, and it remains unclear about the end-to-end efficiency improvements in practice.
- Although the authors clarify that H2A contains multi-step workflows and claim that their prompting supports step-wise tool calling, the core method remains a query-level demonstration retriever, and the rebuttal does not provide strong evidence that it handles stateful multi-turn contexts, and multi-tool selection, which are key aspects raised by the reviewers.

**Reviewer Scores:**

During the rebuttal period, Reviewers ctRF and Hd9m provided responses with positive feedback. For the other reviewers, the following assessment outlines how their scores likely change:

**Reviewer bGHN** (initial rating: 4)

- **W1**: Authors have added two independent judges (GPT-4 and DeepSeek-V3.2) and Wilcoxon significance tests, which have addressed this concern.
- **W2, Q2**: Authors cannot measure true wall-clock due to non-live APIs, and instead provide API call counts. However, the reduction of API calls is marginal, and  the overall improvement in end-to-end efficiency remains unclear.
- **W3, W6, Q3, Q5**: Authors state that H2A already involves multi-turn/multi-tool workflows and demonstrate extension beyond binary behaviors (BAR-Multi). However, parameter selection is still not fully addressed.
- **W4, Q1**: Authors provide a MinHash-based Jaccard similarity to argue improvements are not driven by lexical artifacts.
- **W5, Q6**: Authors attribute regressions to conservative bias in a specific model case, add paired Wilcoxon tests, and provide an $\alpha$ ablation

- **Q4**: Authors report experiments supporting top-5 as best tradeoff and argue stability across benchmarks with same pretrained BAR.

The main concern is still minor efficiency improvements, with only 4.7% on the H2A dataset, and 1% on the ToolDEER dataset. Combined with other evaluations, the reviewer likely maintained a score of 4.

---

**Reviewer Mhtb** (initial rating: 4)

- **W1:** Authors revise Figure 1 with concrete examples to clarify the semantic-behavior clash.
- **W2:** The variable $k$ is explicitly defined in the revised manuscript.
- **W3:** Quantitative correlation analysis and performance trends are added to Section 2.2 to provide the missing evidence for the initial findings.
- **W4:** The authors clarify that cosine similarity is used and that thresholds are determined via standard grid search, addressing the lack of methodological detail.
- **W5:** Authors explain that behavior alignment is achieved through the Dual-Negative Contrastive Loss in the embedding space, which pull similar behaviors together.
- **W6:** The authors incorporate discussions on related works like Confucius and AutoTools to strengthen the paper’s positioning.
- **Q1:** Authors discuss Pseudo Query Generation (PQG) and argue that BAR is complementary because it focuses on behavioral intent rather than just semantic coverage.

Reviewer Mhtb’s primary concern is the unclear presentation of the manuscript. While the authors made several revisions to address these points, the overall presentation still has significant room for improvement. Hence, the reviewer likely maintained a score of 4.

---

### Decision · Program_Chairs · 2026-01-26

Reject